

# Improvement of model evaluation by incorporating prediction and measurement uncertainty

Lei Chen[1], Shuang Li[1], Yucen Zhong[1], Zhenyao Shen[1]

[1] State Key Laboratory of Water Environment Simulation, School of Environment, Beijing Normal University, Beijing 100875, PR China

*Corresponding to:* Zhenyao Shen (zyshen@bnu.edu.cn)

**Abstract.** Numerous research studies have been conducted to assess uncertainty in hydrological and nonpoint source pollution predictions, but few studies have considered both prediction and measurement uncertainty in the model evaluation process. In this study, the Cumulative Distribution Function Approach (CDFA) and the Monte Carlo Approach (MCA) were used to develop two new approaches for model evaluation within an uncertainty framework. For the CDFA, a new distance between the cumulative distribution functions of the predicted data and the measured data was established, whereas the MCA was proposed to address conditions with dispersed data points. These new approaches were then applied in combination with the Soil and Water Assessment Tool in the Three Gorges Region, China. Based on the results, these two new approaches provided more accurate goodness-of-fit indicators for model evaluation compared to traditional methods. The model performance worsened when the error range became larger, and the choice of probability density functions (PDFs) affected model performance, especially for non-point source (NPS) predictions. The case study showed that if the measured error is small and if the distribution can be specified, the CDFA and MCA could be extended to other model applications within an uncertain condition.

**Keywords:** Model evaluation; Uncertainty analysis; Cumulative Distribution Function Approach; Monte Carlo; Soil and Water Assessment Tool; Nonpoint source



pollution

## 1 Introduction

Prediction of non-point source (NPS) pollution has become increasingly utilized
because NPS pollution is a key threat to bodies of water (Shen et al., 2014). Numerous

hydrological models, including the Soil and Water Assessment Tool (SWAT), the
Hydrological Simulation Program-Fortran (HSPF) and the Agricultural Non-Point
Source Model (AGNPS), have been developed and widely applied to hydrological and
NPS (H/NPS) pollution analyses and watershed management (Yang et al., 2008). NPS
pollution is reportedly driven by random and diffuse factors, such as climate, land use,

soil, vegetation cover and human activities (Ouyang et al., 2009), and model confidence
in NPS prediction, represented by model calibration and validation, is currently lacking
in modelling research.

Hydrological models always require input data, optimal parameters and proper model
structure (Baldassarre and Montanari, 2009), whereas data measurement often involves

processes of sampling, transportation and analyses. Errors in these complex processes
lead to uncertainty in the data (Chaney et al., 2015). Uncertainties in hydrology and
NPS modelling are classified as either measurement uncertainty or prediction
uncertainty (Chen et al., 2015; Baldassarre and Montanari, 2009). Uncertainty analysis
is a crucial step in the application of hydrological models (Guinot et al., 2011). The

uncertainty surrounding model structure and parameterization has been extensively
investigated (Wu et al., 2016). Several approaches, including the Generalized
Likelihood Uncertainty Estimation (GLUE) (Hassan et al., 2008; Sathyamoorthy et al.,
2014; Cheng et al., 2014), the Bayesian approach (Freni and Mannina, 2010; Han and
Zheng, 2016; Parkes and Demeritt, 2016; Zhang et al., 2009), Sequential Uncertainty





Fitting (SUFI-2) (Vilaysane et al., 2015; Abbaspour et al., 2007), and Markov Chain

Monte Carlo (MCMC) (Vrugt et al., 2003; Zhang et al., 2016), have been proposed.

However, due to the lack of data, relatively few studies have focused on the inherent

uncertainty in measured data, and even fewer studies have considered measurement and

prediction uncertainties in the evaluation of model performance (Baldassarre and

Montanari, 2009;Montanari and Baldassarre, 2013).

In model evaluation, calibration is the process used to generate optimal parameters for

the best goodness-of-fit between the predicted data and the measured data, and

validation is the process of checking the model performance using another series of

measured data (Chen et al., 2014). Traditional model evaluation only considers the

goodness-of-fit between sets of measured data points and predicted data points

(Westerberg et al., 2011). Such point-to-point methods might be inadequate because

they fail to incorporate the existing uncertainties mentioned above. Previous studies

have noted that if prediction uncertainty exists, the predicted data could be expressed as

a confidence interval (CI) or a probability density function (PDF) (Franz and Hogue,

2011; Shen et al., 2012). Harmel and Smith (2007) used the probable error range (PER)

as an expression of measurement uncertainty and modified the goodness-of-fit

indicators using the deviation term between the predicted data points and the nearest

measurement uncertainty boundaries. Harmel et al. (2010) further modified this

deviation term using a correction factor, which was determined by the degree of overlap

between each paired of measured and predicted intervals. This idea is instructive, but it

might be questionable sometimes because a larger uncertainty or error would result in

more overlap between the prediction and measurement intervals, which would indicate

better model performance. In this regard, Chen et al. (2014) developed an

interval-deviation approach (IDA), which demonstrated that H/WQ models should be





evaluated against both the nearest and farthest boundaries (the inherent uncertainty intervals). Generally, this IDA approach is suitable for incomplete data conditions, but when more data could be collected or when a continuous and random data distribution could be assumed, these intervals may not always be practical. Current research tends

to express uncertain data as certain function distributions to express an error term (Zhang et al., 2009), which might lead to a more feasible expression than either the traditional or IDA methods.

The objective of this study is to develop a new framework for model evaluation by incorporating prediction and measurement uncertainty. Two methods, the Cumulative

Distribution Function Approach (CDFA) and the Monte Carlo Approach (MCA), were proposed for different situations (Sect. 2). Then, the new methods were used in combination with the SWAT to evaluate the Three Gorges Reservoir Area (TGRA), China, as a case study (Sect. 3 and Sect. 4).

## 2 Methodology

In this study, the Nash-Sutcliffe efficiency coefficient (*NSE*) was selected from commonly used indicators, and the expression is as follows:

$$NSE = 1 - \sum_{i=1}^{N}(O_i - P_i)^2 / \sum_{i=1}^{N}(O_i - \overline{O}_i)^2 \tag{4}$$

where $\{O_i| i = 1, 2,. . .,N\}$ is the set of measured data, $\{P_i| i = 1, 2,. . .,N\}$ is the set of predicted data and $\overline{O}$ is the mean value of the measured data.

In traditional indicators, the deviation between the measured and predicted data is expressed by the absolute distance $(O_i - P_i)$ between the paired data points. This method is questionable because it fails to incorporate prediction and measurement uncertainty. In this paper, the probability distributions of each data set were statistically estimated, and the calculations of $O_i - P_i$ were modified by using stochastic distances

between the paired PDFs. For the CDFA, cumulative distribution functions were used to describe uncertain data because they are simple and do not depend on the distributional properties throughout the data sets (see Sect. 2.2). A topological distance, which is based on the distance between cumulative distribution functions

(distribution-to-distribution), was proposed to replace the traditional error item in the model evaluation. The Monte Carlo method was also used to generate groups of discrete uncertain data throughout the sampling process (Vrugt and Ter Braak, 2011). Thus, the MCA was proposed as a supplement to the CDFA when the uncertain data were discrete or when no specific distributions could be used (see Sect. 2.2). A

flowchart of the model evaluation within the uncertainty framework is presented in Figure 1.

## 2.1 The description of the CDFA method

The idea behind the CDFA was to replace the point-to-point comparison with the deviation between uncertain measured data and predicted data expressed as cumulative

distribution functions. A topological distance was proposed, and visualizations of the topological distance are illustrated in Figure 2. The cumulative distribution function was chosen because it is a monotone increasing function with a limited threshold and an integral property. The distance between cumulative distributions was then transformed into an area topological distance (D). The proof of the rationality behind the topological

distance D is shown in Table S1 (in the supplementary material).

Based on Table S1, (B, D) is a reasonable metric space, and D is as a reasonable measurement of B. Therefore, the difference between $F_p(x)$ and $F_o(x)$ is reasonable and advisable.

The detailed steps of the CDFA are as follows:



1) The prediction and measurement uncertainty are generated using GLUE, PER or other methods;

2) The prediction and measurement data intervals are analysed, and the cumulative distribution functions of the prediction uncertainty ($F_o(x)$) and the measurement uncertainty ($F_p(x)$) are calculated.

3) The topological interval (area distance) between the two functions $F_o(x)$ and $F_p(x)$ is quantified.

4) The new $O_i - P_i$ is quantified, and the modified evaluation indicators are used for model evaluation.

**2.2 The MCA method**

In other cases, the measurement and prediction uncertainties might be expressed as discrete data, or no continuous distribution function may fit the data set. For example, the input uncertainty relating to spatial rainfall variability might only result in a limited number of predicted data points that cannot be expressed as certain PDFs (Shen et al., 2012). To incorporate this type of uncertainty, MCA was implemented using the Monte Carlo technique, which has been used in many hydrological uncertainty studies (Sun et al., 2008; Zhang et al., 2016). The Monte Carlo technique is a type of random sampling method that considers combinations of different input components and determines a statistical distribution for the output data (Shen et al., 2013). A key step is sampling variables randomly for discrete data so that the measurement and prediction data can be expressed as certain distributions. Here, ($O_i - P_i$) was replaced by a stochastic expression of the deviation between pairs of data groups, and these stochastic deviations were then used to calculate the evaluation indicators. The details of the MCA are as follows:





1) The distribution functions or discrete measured data points ($f_o(x)$) and predicted data ($f_p(x)$) are generated.

2) The sampling process of $f_p(x)$) and $f_o(x)$ is realized using the Latin Hypercube Sampling approach (Shen et al., 2012), and the software Crystal Globe was used to sample for the MCA.

3) Based on the random samples of the predicted and measured data, corresponding individual goodness-of-fit indicators are calculated.

4) The sampling process is repeated until the target sample size is achieved.

5) A group of goodness-of-fit indicator values are obtained, and these values are used to produce the statistical analysis for the model evaluation within the uncertainty framework.

## 3 Case study

In this study, the Daning Watershed (108 °440–110 °110E, 31 °040–31 °440N), which is located in the central part of the TGRA, was selected as the study area. Previously, the uncertainty ranges related to the flow, sediment and TP predictions were quantified using the GLUE method (Chen et al., 2014; Shen et al., 2012), and these results and uncertainty ranges were used as the predicted data sets. Normal, uniform and lognormal distributions, which are classic and simple PDFs, were assumed for each predicted data set. More details about the uncertainty ranges and PDFs of the predicted flow, sediment and total phosphorus (TP) can be found in our previous study (Chen et al., 2014).

The measured stream flow, sediment and TP data at the Wuxi hydrological gauges were obtained from the Changjiang Water Resources Commission. Due to data limitations, the error range of the measured data was derived from Harmel et al. (2006), Harmel and Smith, (2007) and Harmel et al. (2010). Based on our previous study (Chen et



al.,2014), the measurement uncertainty was assumed to be a normal distribution in this paper, and three scenarios, an ideal case, a typical case and a worst case, were used. The probable error ranges (PERs) for flow, sediment and TP were 2%, 2% and 2%, respectively, for each ideal case scenario; 9%, 16% and 26%, respectively, for each

typical case scenario; and 36%, 102% and 221%, respectively, for each worst-case scenario.

## 4 Results and discussion

### 4.1 The model evaluation results using the CDFA

The model evaluation results for flow, sediment and TP are shown in Table 1. For

simplicity, only the *NSE* indicator was chosen as a model evaluation indicator, and the model evaluation results using a traditional point-to-point method were used as a baseline scenario. For the traditional method, the *NSE* values were 0.736, 0.642 and 0.783 for flow, sediment and TP, respectively. Using the CDFA method (assuming the measured error was small (the ideal case)), the following changes to the *NSE* values

were obtained: 0.752, 0.660 and 0.810 for flow, sediment and TP, respectively, in the normal distribution scenario; 0.742, 0.661 and 0.814, respectively, in the uniform distribution scenario; and 0.752, 0.660 and 0.812, respectively, in the lognormal distribution scenario. However, when the measurement error became large (for the typical case), the following *NSE* values were obtained: 0.751, 0.657 and 0.789 for flow,

sediment and TP, respectively, in the normal distribution scenario; 0.742, 0.661 and 0.814, respectively, in the uniform distribution scenario; and 0.751, 0.657 and 0.791, respectively, in the lognormal distribution scenario. When the measurement error became negative (the worst-case scenario), the following *NSE* values were obtained: 0.744, 0.551 and -0.056 for flow, sediment and TP, respectively, in the normal

distribution scenario; 0.736, 0.545 and -0.019, respectively, in the uniform distribution





scenario; and 0.744, 0.437 and -0.072, respectively, in the lognormal distribution scenario.

### 4.2 The model evaluation results using MCA

The sampling size is important for MCA, so a sensitivity analysis was first conducted.

Groups of $O_i$ and $P_i$ values (10, 50, 100, 200, 500, 1000, 2000 and 5000) were randomly generated and used to calculate the *NSE*, and the sampling sizes were obtained using statistical analysis of the *NSE* (only the results for 1000, 2000, and 5000 are shown in Table 2). The sampling results showed that with increasing sampling size, the mean value and the coefficients of variation ($C_V$) of the flow, sediment and TP also

increased. However, when the sampling sizes are larger than 2000, the model performance become stable, and all indicators only changed within 1%, indicating that larger sampling sizes of $O_i$ and $P_i$ would not further benefit the performance of the model. Thus, a sampling size of 5000 was chosen in this study.

The evaluation results, which are expressed as the 95% confidence interval of the *NSE*

for the flow, sediment and TP predictions, are shown in Table 1. The *NSE* ranges for flow, sediment and TP in the ideal case were as follows: 0.73~0.74, 0.61~0.69 and 0.71~0.82, respectively (normal distribution); 0.73~0.75, 0.60~0.70 and 0.71~0.84, respectively (uniform distribution); and 0.73~0.74, 0.61~0.69 and 0.69~0.81, respectively (lognormal distribution). The *NSE* ranges for flow, sediment and TP in the

typical case were as follows: 0.71~0.75, 0.59~0.69 and 0.62~0.83, respectively (normal distribution); 0.71~0.76, 0.59~0.71 and 0.55~0.86, respectively (uniform distribution); and 0.71~0.75, 0.59~0.68 and 0.62~0.83, respectively (lognormal distribution). The *NSE* ranges for flow, sediment and TP in the worst case were as follows: 0.63~0.79, -0.31~0.67 and -3.10~0.67, respectively (normal distribution); 0.63~0.79, -0.53~0.68





and -3.27~0.72, respectively (uniform distribution); and 0.63~0.79, -0.28~0.66 and

-3.01~0.67, respectively (lognormal distribution).

### 4.3 Analysis of influencing factors

### 4.3.1 Impact of error range

Generally, the data uncertainty range should always be obtained by analysing a large

amount of data, so it is difficult to ensure the error range of the predicted or measured

data due to data limitations. In this study, the measurement error is expressed as the

PER, and three PERs were obtained as expressions of different error ranges (Harmel

and Smith, 2007). In this section, the error ranges of the measured data were assumed as

the PERs, and the impacts of the PERs on the evaluation results of the CDFA and the

MCA were quantified. Only the normal distribution was considered for the prediction

data. For the ideal case scenario (PER of 2%), the $NSE$ for the flow evaluation was

0.752, but for the typical case and the worst-case scenarios, the values of the $NSE$

changed to 0.751 and 0.744, respectively. Compared to the point-to-point result, the

goodness-of-fit indicators obtained from the CDFA ($NSE$) increased by 21.3% for flow

in the ideal case. The $NSE$ increased by 20.1% and 9.8% for the typical case and the

worst-case scenarios, respectively. Similar variation in the evaluation results were

observed for the MCA method. The $NSE$ for the flow evaluation was 63.5% for the

ideal case (normal distribution) and was 40.9% and 10.6% for the typical case and the

worst case, respectively (Harmel and Smith, 2007; Shen et al., 2013b). For flow

prediction, the evaluation results obtained using the CDFA were all satisfactory with

measurement error of any size; however, for the sediment and TP evaluations, the

goodness-of-fit indicators became unacceptable if the measurement errors were large

(in the worst-case scenario). In this regard, the range of measurement error showed

different impacts on the flow, sediment and TP predictions. For example, the $NSE$



values were 0.752, 0.660 and 0.810 for the flow, sediment and TP evaluations in the ideal case scenario. From the results above, a large measurement error would cause decreasing evaluation performance, which is different from the results of Harmel and Smith (2007). Similar results were observed for the MCA.

As shown in Table 1, increasing measurement error would lead to decreased *NSE*, which means less confidence in the model performance. The worst evaluation indicators were observed when the measurement error was the largest. The performance of the TP predictions became unacceptable when the PER was 221% (worst case). This result indicated that a threshold error range might exist for model

evaluation. When the error range is less than this specific value (such as the ideal and typical cases for TP prediction used in this study), the model evaluation result is acceptable. However, if the measurement error exceeds this threshold value (worst case), the model evaluation would be unacceptable, and the confidence in the model performance would be lost, especially for the NPS prediction. However, in reality, it is

often difficult to accurately measure pollutant data, especially in developing countries, such as China. In these countries, a "calibrated model" would have few advantages over an un-calibrated model because of the lack of precisely measured data.

### 4.3.2 Impacts of data distribution

The assignment of PDFs might be the most difficult and subjective task in the

application of uncertainty analysis to hydrological models (Shen et al., 2013). Thus, model performance might be influenced not only by the error range but also by the choice of PDF. In this study, the prediction uncertainty was assumed as three certain distribution functions, but it is always difficult to ensure which PDFs should be used (Shen et al., 2013). In this section, we further quantified the impacts of the different

PDFs on model performance. For simplicity, only the results of the CDFA and typical



case scenario are considered here. For the flow evaluation, the *NSE* values were 0.751, 0.742 and 0.751 when the predicted data were modelled using the normal distribution, uniform distribution and lognormal distribution, respectively. For TP evaluation, the *NSE* values were 0.789, 0.814 and 0.791 for the normal distribution, uniform distribution and lognormal distribution, respectively. As in a previous study, the CI of the prediction was larger for flow than for TP (Shen et al., 2012). Thus, the prediction distributions have a low impact on the evaluation in cases of high CI values but have a bigger impact on the evaluation when the CI of the prediction is low. Compared to the baseline scenario, the *NSE* values for the hydrological prediction increased by 20.1%, 7.3% and 20.3% for the normal, uniform and lognormal distributions, respectively. These results indicated that the choice of PDF would show certain impacts the model evaluation for hydrological, sediment and TP applications. This result is consistent with previous studies, which also showed that prediction uncertainty distributions can affect the goodness-of-fit indicators (Harmel et al., 2010). Table 1 also indicates that the choice of predicted PDFs should be dependent on the selection of the measured PDFs. If the measurement and prediction uncertainties are set using the same PDFs, such as a normal distribution, the goodness-of-fit indicators would be larger, indicating a more reliable model performance. Thus, the choice of proper PDFs is important to make accurate model evaluation for NPS predictions. Based on these results, we suggest that model acceptability can be attained by using certain PDFs on the model output by collecting information from model documentation, previous studies, and other literature to make an "educated guess".

### 4.4 Comparison with previous methods

In a previous study, Harmel and Smith (2007) advanced the IDA method, and this "point-to-interval" method was based on the distance between the nearest boundaries of



paired intervals. Compared to our results, the difference between the paired data intervals or the paired PDFs overlapping for larger uncertainties would be mistakenly regarded as "no difference". The point-to-interval method gave a higher goodness-of-fit, but the measured data were only treated as data points. In this regard, deviations

between measurements and the prediction data would be ignored in the model evaluation, which is not appropriate because the measurement error range would greatly affect the model performance (as mentioned in section 4.3).

Chen et al. (2014) improved the nearest method by correcting for the overlapping parts of the uncertainty data and using both the nearest and farthest boundaries. Using the

IDA, the *NSE* for the hydrological prediction would be 0.834 in the ideal case. However, the CDFA method produced lower *NSE* values, which were 0.752 for the normal distribution, 0.742 for the uniform distribution and 0.752 for the lognormal distribution. In the typical case, the IDA method would produce an *NSE* value of 0.833, but the CDFA would result in an *NSE* value of 0.751 for the normal distribution, 0.742 for the

uniform distribution, and 0.751 for the lognormal distribution. The difference between the IDA and the CDFA would be largest for the worst case, in which the *NSE* values would be 0.780 for the IDA method and 0.744 (normal distribution), 0.736 (uniform distribution), or 0.744 (lognormal distribution) for the CDFA method (all results of the IDA can be found in Chen et al., 2014).

In Chen et al. (2014), an "interval-to-interval" method was proposed in which an absolute distance between measurement and prediction uncertainty data was derived from both the nearest and farthest boundaries. However, due to data limitations, a weight factor was used to balance the nearest boundaries and the farthest boundaries, and the choice of weight factor was subjective. When the weight factor was set to 0.5,

the IDA method would produce similar goodness-of-fit indicators as the results of the





CDFA using the uniform or normal distributions for both the predicted and measured data (Chen et al., 2014). For example, the *NSE* value for the hydrological prediction was 0.764 using the IDA method; if the CDFA was used, the goodness-of-fit indicators would be 0.752, 0.742 and 0.752 for the normal, lognormal, and uniform distributions, respectively. Therefore, when specific PDFs were used, the IDA method could be viewed as a simplification of the CDFA. Previous studies have also indicated that the lognormal distribution provides a relatively close approximation to the true error characteristics, so the CDFA could be more practical if certain prediction uncertainties exist (Shen et al., 2015).

**5 Conclusion**

In this study, two new methods were proposed and employed to evaluate model performance within an uncertainty framework: the CDFA and the MCA. Using the CDFA and the MCA, both prediction and measurement uncertainty were considered, and the possible impacts of error range and the choice of PDFs were quantified for a real application. Based on the results, the model performance worsened when a larger error range existed, and the choice of PDF affected the model performance, especially for NPS predictions. These proposed methods could be extended to other goodness-of-fit indictors and other watershed models to provide a substitution for traditional model evaluations within an uncertainty framework.

With the results presented, modellers should better assess the error range of measured data for their use in watershed simulations, and more data should be gathered to obtain a real measurement error range and a proper PDF for the predicted data. Further explanations are also suggested for the inherent uncertainty of hydrological and



pollutant transportation processes. More case studies should be conducted to test the IDA, CDFA and MCA in future practical analyses of other watershed models.

*Acknowledgement.* This research was funded by the National Natural Science Foundation of China (No. 51579011, & 51409003) and the Fund for Innovative

Research Group of the National Natural Science Foundation of China (No. 51421065).

*Competing interests.* The authors declare that they have no conflict of interest.

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





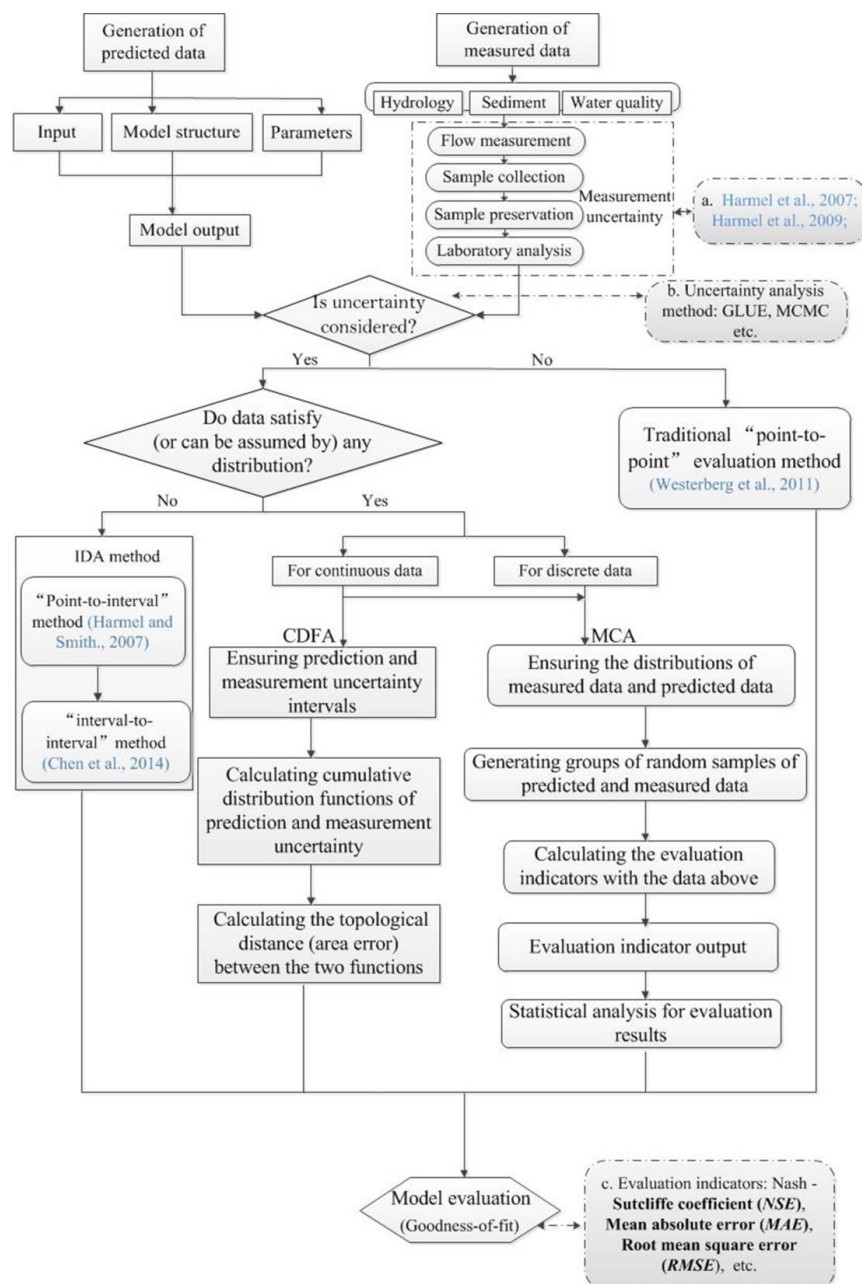

**Figure 1.** A general flow chart of model evaluation within the uncertainty framework.





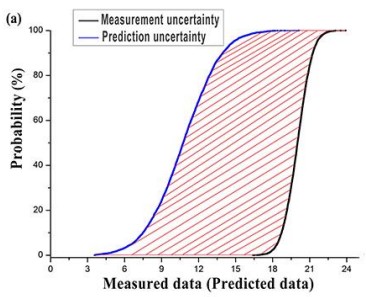
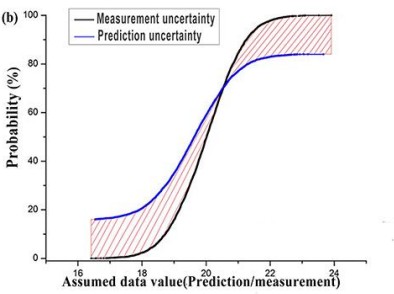

**Figure 2.** Expression of topological distance for (a) the case in which the measured and predicted data are non-overlapping and (b) the case in which the measured and predicted data are overlapping.





**Table 1.** The goodness-of-fit indicators for hydrological, sediment and TP models using both the new and traditional methods.

| Variable | Indicator | Point-to-point method | Normal distribution | | | Uniform distribution | | | Lognormal distribution | | |
|---|---|---|---|---|---|---|---|---|---|---|---|
| | | | Ideal | Typical | Worst | Ideal | Typical | Worst | Ideal | Typical | Worst |
| Flow | NSE (CDFA) | 0.736 | 0.752 | 0.751 | 0.744 | 0.742 | 0.742 | 0.736 | 0.752 | 0.751 | 0.744 |
| | NSE (MCA) | | 0.73-0.74 | 0.71-0.75 | 0.63-0.79 | 0.73-0.75 | 0.71-0.76 | 0.63-0.79 | 0.73-0.74 | 0.71-0.75 | 0.63-0.79 |
| Sediment | NSE (CDFA) | 0.642 | 0.660 | 0.657 | 0.551 | 0.661 | 0.661 | 0.545 | 0.660 | 0.657 | 0.437 |
| | NSE (MCA) | | 0.610-0.690 | 0.59-0.69 | -0.31-0.67 | 0.600-0.700 | 0.59-0.71 | -0.53-0.68 | 0.61-0.69 | 0.59-0.68 | -0.28-0.66 |
| TP | NSE (CDFA) | 0.783 | 0.810 | 0.789 | -0.056 | 0.814 | 0.814 | -0.019 | 0.812 | 0.791 | -0.072 |
| | NSE (MCA) | | 0.71-0.82 | 0.62-0.83 | -3.10-0.67 | 0.71-0.84 | 0.55-0.86 | -3.27-0.72 | 0.69-0.81 | 0.62-0.83 | -3.01-0.67 |





**Table 2.** The result of sampling (2000 times and 5000 times) the flow, sediment and TP in different distributions.

| | Number of simulations | Normal distribution | | Uniform distribution | | Lognormal distribution | |
|---|---|---|---|---|---|---|---|
| | | $M_V^a$ | $C_V^b$ | $M_V$ | $C_V$ | $M_V$ | $C_V$ |
| Flow — Ideal | 1000 | 0.738 | 0.005 | 0.742 | 0.006 | 0.736 | 0.005 |
| | 2000 | 0.737 | 0.005 | 0.742 | 0.006 | 0.736 | 0.005 |
| | 5000 | 0.737 | 0.005 | 0.742 | 0.006 | 0.736 | 0.005 |
| Flow — Typical | 1000 | 0.734 | 0.013 | 0.736 | 0.013 | 0.735 | 0.014 |
| | 2000 | 0.734 | 0.013 | 0.738 | 0.013 | 0.732 | 0.013 |
| | 5000 | 0.733 | 0.013 | 0.737 | 0.013 | 0.733 | 0.013 |
| Flow — Worst | 1000 | 0.578 | 0.055 | 0.824 | 0.014 | 0.713 | 0.015 |
| | 2000 | 0.683 | 0.047 | 0.748 | 0.013 | 0.730 | 0.013 |
| | 5000 | 0.693 | 0.048 | 0.737 | 0.013 | 0.737 | 0.013 |
| Sediment — Ideal | 1000 | 0.643 | 0.029 | 0.657 | 0.043 | 0.641 | 0.028 |
| | 2000 | 0.642 | 0.030 | 0.657 | 0.044 | 0.642 | 0.028 |
| | 5000 | 0.642 | 0.030 | 0.657 | 0.044 | 0.642 | 0.029 |
| Sediment — Typical | 1000 | 0.639 | 0.039 | 0.652 | 0.05 | 0.64 | 0.057 |
| | 2000 | 0.640 | 0.038 | 0.654 | 0.049 | 0.637 | 0.038 |
| | 5000 | 0.638 | 0.039 | 0.653 | 0.049 | 0.638 | 0.038 |
| Sediment — Worst | 1000 | 0.378 | 0.818 | 0.484 | 0.987 | 0.467 | 1.224 |
| | 2000 | 0.440 | 0.713 | 0.440 | 0.906 | 0.415 | 1.21 |
| | 5000 | 0.446 | 0.717 | 0.433 | 0.914 | 0.424 | 1.213 |
| TP — Ideal | 1000 | 0.773 | 0.038 | 0.782 | 0.044 | 0.771 | 0.039 |
| | 2000 | 0.772 | 0.039 | 0.782 | 0.044 | 0.771 | 0.039 |
| | 5000 | 0.771 | 0.039 | 0.781 | 0.045 | 0.772 | 0.040 |
| TP — Typical | 1000 | 0.744 | 0.074 | 0.747 | 0.106 | 0.747 | 0.106 |
| | 2000 | 0.745 | 0.073 | 0.749 | 0.104 | 0.744 | 0.072 |
| | 5000 | 0.743 | 0.073 | 0.748 | 0.105 | 0.746 | 0.072 |



| Worst | | | | | | |
|---|---|---|---|---|---|---|
| 1000 | -0.091 | -12.224 | -0.168 | -8.907 | -0.172 | -8.529 |
| 2000 | -0.106 | -10.543 | -0.153 | -8.172 | -0.148 | -8.737 |
| 5000 | -0.108 | -10.716 | -0.150 | -8.249 | -0.156 | -8.449 |

ᵃ $M_V$ is the mean value; ᵇ $C_V$ is the coefficient of variation.

