# Peer review of "Improvement of model evaluation by incorporating prediction and measurement uncertainty"

_Hydrology and Earth System Sciences, 2017_

## Referee Comment (RC1) · Anonymous Referee #1 · 7 Oct 2017

Uncertainty analysis constitute important challenge in hydrological and water quality modeling area. In this manuscript, the authors proposed a new approach to improve model evaluation by incorporating prediction and measurement uncertainty. This new method, if proven as valid, will contribute to the development of uncertainty analysis techniques and interest audience of the journal substantially. However, after reading this manuscript, I have some concerns, which are listed below:

About the CDFA method:

The idea of evaluating the goodness of model fit by comparing the distributions of model predicted values and observed values is appealing. However, in typical setting of a hydrological/water quality modeling problem, the distributions of model predictions are constructed using certain "calibration" process, as already indicated in manuscript.

[Figure]

The definition of measure of goodness of fit, which could be used for model evaluation such as the Nash–Sutcliffe model efficiency, should be able to be used to help reduce the parametric uncertainty of the model or, using the terms from GLUE, differentiate between "behavioral" and "non-behavioral" parameter sets. I did not see the CDFA approach proposed in this manuscript can function this way according to the description of the method provided in section 2.1. To put it another way, when there is knowledge of uncertainty or distributions of observed values, it is desirable to incorporate this knowledge into model calibration. However, it seems to me that the proposed CDFA approach does not provide a way to allow that information into model calibration and simply provides an alternative metric to summarize the model calibration results at the post-calibration stage. The utility of the new approach is therefore not significant.

About MCA approach:

As for the MCA approach, I am afraid I could not find which variables are discrete variables of interest in the case study designed to demonstrate the implementation of MCA approach (section 4.2). All SWAT output variables mentioned in the case studies seem to be continuous.

Of course, I would like to acknowledge that providing a thorough review on theoretical work in uncertainty analysis such as the one presented in this manuscript could be quite challenging. The concerns noted above may be due to my limited knowledge. But I hope that the authors can provide an explanation and clarification to these problems.

---

## Referee Comment (RC2) · Anonymous Referee #2 · 16 Nov 2017

Authors propose new approaches based on cumulative distribution functions of the predicted and observed data (CDFA) and the Monte Carlo approach in combination with the Soil and Water Assessment Tool (SWAT) to assess model performances within an uncertainty framework in the Three Gorges Region, China, . They reported that the proposed approaches perform better than the classical goodness of fits criteria and that the proposed methods could be extended to other goodness-of-fit indictors and other watershed models to provide a substitution for traditional model evaluations within an uncertainty framework.

The idea of combining both predictive and observed uncertainty to assess model performances and uncertainty is quite interesting. However, it is not clear for me how the assessed uncertainty is used with the model to gain knowledge and to improve model

performance? then, how can the proposed approaches be implemented within the calibration process to reduce model error?

---

## Author Response (AR1)

Dear Editor,

Thank you very much for your suggestions. We have carefully considered your comments and revised the manuscript. And more detailed description about CDFA could be founded in **Response to the Reviewer 1 and Reviewer 2**. And we would like to clearly descript the novelty, effectiveness, and limitations of the CDFA method proposed as following:

The objective of this study is indeed to develop a new framework for model evaluation by incorporating prediction and measurement uncertainty. In traditional indicators (such as Nash–Sutcliffe model efficiency), the deviation between the measured and predicted data is expressed by the absolute distance $(O_i - P_i)$ between the paired data points. This method is questionable because it fails to incorporate prediction and measurement uncertainty. Thus, the idea behind the CDFA was to replace the point-to-point comparison with the deviation between uncertain measured data and predicted data expressed as cumulative distribution functions. In fact, this is a modification of traditional good-of-fit indicators by replacing the calculations of their $O_i - P_i$ term by using stochastic distances between the paired probability density functions (PDFs). Thus, this CDFA could be used during the calibration and validation process if PDFs could be obtained for both prediction data and measurement data. Based on the results obtained from this study, we found that the model performance worsened when a larger error range existed, and the choice of PDF affected the model performance, especially for non-point source (NPS) pollution predictions. These proposed methods could be extended to other goodness-of-fit indictors and other watershed models to provide a substitution for traditional model evaluations within an uncertainty framework. Thus, the authors do believe our method could be a substitute of traditional goodness-of-fit indictors and they could be used for the calibration and validation process.

However, we only compare the results between traditional Nash–Sutcliffe model efficiency and CDFA during certain "post-calibration" process. The reasons could be due to the imperfect knowledge of prediction uncertainty as well as the huge calculation effects. We could not set fixed PDFs or error range for prediction data due to insufficient knowledge and natural randomness. Besides, the CDFA is more complex than the traditional indicators so this method would take more running time. Although the increasing techniques expand the calibration process, the execution of CDFA is computationally expensive and technical complex, especially for large numbers of simulations during the parameter optimization process.

There are a number of concerns surrounding the application of this new algorithm in practice. In fact, we have focused on this issue (using the CDFA for model calibration) recently. First, we have considered different sources of uncertainty and the possible range of model performances in a real application. Second, we have incorporated a more realistic but simpler simulator to decrease the computational complexity while making full use of CDFA's strengths. Thus, these issues would be solved in the future, and CDFA could be applied effectively for model calibration.

**Response to the Reviewer 1**

**Comments: 1)** The idea of evaluating the goodness of model fit by comparing the distributions of model predicted values and observed values is appealing. However, in typical setting of a hydrological/water quality modeling problem, the distributions of model predictions are constructed using certain "calibration" process, as already indicated in manuscript. The definition of measure of goodness of fit, which could be used for model evaluation such as the Nash–Sutcliffe model efficiency, should be able to be used to help reduce the parametric uncertainty of the model or, using the terms from GLUE, differentiate between "behavioral" and "non-behavioral" parameter sets. I did not see the CDFA approach proposed in this manuscript can function this way according to the description of the method provided in section 2.1. To put it another way, when there is knowledge of uncertainty or distributions of observed values, it is desirable to incorporate this knowledge into model calibration. However, it seems to me that the proposed CDFA approach does not provide a way to allow that information into model calibration and simply provides an alternative metric to summarize the model calibration results at the post-calibration stage. The utility of the new approach is therefore not significant.

**Response**: Thank for very much for this suggestion. I agree with your idea that the distributions of model predictions and measurement data should be used for the calibration process. In fact, the objective of this study is indeed to develop a new framework for model evaluation by incorporating prediction and measurement uncertainty. In traditional indicators (such as Nash–Sutcliffe model efficiency), the deviation between the measured and predicted data is expressed by the absolute distance $(O_i - P_i)$ between the paired data points. This method is questionable because it fails to incorporate prediction and measurement uncertainty. In this regard, we have developed an interval-deviation approach (IDA), which demonstrated that H/WQ models should be evaluated against both the nearest and farthest boundaries (the

inherent uncertainty intervals). And this work has been published in Chen et al. (2014). However, we have noted that this IDA approach is suitable for incomplete data conditions, but when more data could be collected or when a continuous and random data distribution could be assumed, these intervals may not always be practical. Thus, we have developed CDFA method.

The idea behind the CDFA was to replace the point-to-point comparison with the deviation between uncertain measured data and predicted data expressed as cumulative distribution functions. In fact, this is a modification of traditional good-of-fit indicators by replacing the calculations of their $O_i - P_i$ term by using stochastic distances between the paired probability density functions (PDFs). Thus, this CDFA could be used during the calibration and validation process if PDFs could be obtained for both prediction data and measurement data (why we only did it during post-calibration stage would be explained below). This process could be described as below:

1) The prediction and measurement uncertainty are generated or assumed using previous knowledge.

2) The prediction and measurement data intervals are analysed, and the cumulative distribution functions of the prediction uncertainty ($F_o(x)$) and the measurement uncertainty ($F_p(x)$) are calculated.

3) The topological interval (area distance) between the two functions $F_o(x)$ and $F_p(x)$ is quantified.

4) The new $O_i - P_i$ is quantified, and the modified evaluation indicators are used for model evaluation.

Based on the results obtained from this study, we found that the model performance worsened when a larger error range existed, and the choice of PDF affected the model performance, especially for non-point source (NPS) pollution predictions. These proposed methods could be extended to other goodness-of-fit indictors and other watershed models to provide a substitution for traditional model evaluations within an uncertainty framework. Thus, the authors do believe our method could be a substitute of traditional goodness-of-fit indictors and they could be used for the calibration and validation process.

However, it should be noted that we only compare the results between traditional Nash–Sutcliffe model efficiency and CDFA during certain "post-calibration" process. The reasons could be due to the imperfect knowledge of prediction uncertainty as well

as the huge calculation effects. First, the meteorological-, geological-, hydrological-, and ecological processes in catchments are notably complex and are not always well known. Faced with such insufficient knowledge and natural randomness, uncertainty becomes an inherent part of watershed modeling. On one hand, measurement uncertainty may stem from errors in flow measurements, water quality sample collection, the processes of preservation, storage, transport and laboratory analysis, could be fixed as certain PDFs. In a thorough review (Harmel et al., 2006), all possible errors in the H/WQ measured data were compiled, indicating that appreciable inherent errors exist in the measured data even when following strict quality assurance and quality control (QA/QC) guidelines (Beven et al., 2012). Thus, we could fix certain possible errors or PDFs for each H/WQ measured data and used them during the calibration or validation process. One the other hand, the prediction uncertainty is more complex due to different sources of uncertainty, uncertainty propagation, evaluation methods, uncertainty expression and the control of uncertainty. Beck (1987) reported that residual uncertainty exists even with the best model structure and input data. Thus, we could not set fixed PDFs or error range for prediction data due to insufficient knowledge and natural randomness. That is the main reason why we did not use the CDFA method during the whole calibration process. Second, it should be noted the CDFA is more complex than the traditional indicators so this method would take more running time. Although the increasing techniques expand the calibration process, the execution of CDFA (also for MCA) is computationally expensive and technical complex, especially for large numbers of simulations during the parameter optimization process. In this paper, we have tried the CDFA into the calibration process (only for CN2) but we found the computing time is large.

We agree with your idea that there are a number of concerns surrounding the application of this new algorithm in practice. In fact, we have focused on this issue (using the CDFA for model calibration) recently. First, we have considered different sources of uncertainty and the possible range of model performances in a real application. Second, we have incorporated a more realistic but simpler simulator to decrease the computational complexity while making full use of CDFA's strengths. Thus, these issues would be solved in the future, and CDFA could be applied effectively for model calibration.

**Comments: 2)** As for the MCA approach, I am afraid I could not find which variables are discrete variables of interest in the case study designed to demonstrate the

implementation of MCA approach (section 4.2). All SWAT output variables mentioned in the case studies seem to be continuous.

**Response:** In this study, two methods, the Cumulative Distribution Function Approach (CDFA) and the Monte Carlo Approach (MCA), were proposed for different situations. For the CDFA, cumulative distribution functions were used to describe uncertain data because they are simple and do not depend on the distributional properties throughout the data sets. The MCA was proposed as a supplement to the CDFA when the uncertain data were discrete or when no specific distributions could be used. A flowchart of the model evaluation within the uncertainty framework is presented in Figure 1.

Previous studies have noted that if prediction uncertainty exists, the predicted data could be expressed as a confidence interval (CI) or a probability density function (PDF) (Franz and Hogue, 2011; Shen et al., 2012). Current research tends to express uncertain data as certain function distributions to express an error term (Zhang et al., 2009), which might lead to a more feasible expression than the traditional indicators. However, prediction uncertainty can be expressed as discrete variables of interest. For example, uncertainty related to rainfall is currently recognized as the major challenge for hydrological modeling science. Many previous studies have investigated uncertainty associated with measurement errors and spatial variability associated with rainfall. A number of researchers have investigated deviations in measured data, such as Sun et al. (2000), Kavetski et al. (2006), Bárdossy and Das (2008), McMillan et al. (2011). This uncertainty originates mainly from inaccuracy in measuring devices, local meteorological effects, and errors in data transmission. This kind of uncertainty could be expressed as confidence interval (CI) or a probability density function (PDF).

However, another important source of uncertainty that can be expressed as discrete variables also existed. For example, even in ideal conditions, where the dense and well-distributed gauges are available, the rain-gauge network cannot fully capture every point over the watershed. It is more common to have only a few stations distributed in space over the watershed. Rainfall at unknown points is thus estimated by means of interpolation techniques. Several techniques—such as the Centroid method, the Thiessen Polygon method, IDW and the Kriging method—have been used in spatial interpolation to produce information on the spatial distribution of rainfall (Mamillapalli, 1998; Chaubey et al., 1999; Bárdossy and Das, 2008; Hamed et al., 2009; Cho et al., 2009; Fu et al., 2011). It is therefore logical to take interpolation

methods into account when determining the impacts of spatial rainfall variability on H/NPS predictions in large watershed.

We have investigated variability in spatial rainfall estimates associated with interpolation methods on modeling in large watershed. In our previous studies, the uncertainty introduced by spatial rainfall variability was determined using a number of different interpolation methods. These comprehensively-used techniques are: 1) the Centroid method; 2) the Thiessen Polygon method; 3) the Inverse Distance Weighted (IDW) method; 4) the Disjunctive Kriging method, and 5) the Co-Kriging method. A semi-distributed model—the Soil and Water Assessment tool (SWAT) was used in a large watershed in the Three Gorges Reservoir Area (TGRA), China. The modeling outputs considered were flow, sediment, and total phosphorus (TP) at the watershed outlet. Results indicated that spatial interpolation techniques resulted in considerable uncertainty of rainfall spatial variability and transferred even larger uncertainty to H/NPS modeling. Similar studies could be also been found in our other previous studies. For example, we have been carried out into the effect of GIS data on water quality modeling and the uncertainty related to the combination of the available GIS maps (Shen et al., 2013). Besides, we have focused on the structural uncertainty caused by the algorithms and equations that are used to describe the phosphorus (P) cycle at the watershed scale. All these kinds of prediction uncertainty relating to limited model structures or model input datasets could result in discrete variables.

Thus, we also considered this kind of the measurement and prediction uncertainties, which might be expressed as discrete data. To incorporate this type of uncertainty, MCA was implemented using the Monte Carlo technique, which has been used in many hydrological uncertainty studies (Sun et al., 2008; Zhang et al., 2016). The Monte Carlo technique is a type of random sampling method that considers combinations of different input components and determines a statistical distribution for the output data (Shen et al., 2013). A key step is sampling variables randomly for discrete data so that the measurement and prediction data can be expressed as certain distributions. Here, $(O_i - P_i)$ was replaced by a stochastic expression of the deviation between pairs of data groups, and these stochastic deviations were then used to calculate the evaluation indicators.

**Response to the Reviewer 1**

**Comments: 1)** The idea of combining both predictive and observed uncertainty to assess model performances and uncertainty is quite interesting. However, it is not

clear for me how the assessed uncertainty is used with the model to gain knowledge and to improve model performance?

**Response**: Thank for very much for this suggestion. In fact, the objective of this study is indeed to develop a new framework for improving the evaluation of model performance by incorporating prediction and measurement uncertainty. In traditional indicators (such as Nash–Sutcliffe model efficiency), the deviation between the measured and predicted data is expressed by the absolute distance ($O_i - P_i$) between the paired data points. This method is questionable because it fails to incorporate prediction and measurement uncertainty. In this regard, we have developed an interval-deviation approach (IDA), which demonstrated that H/WQ models should be evaluated against both the nearest and farthest boundaries (the inherent uncertainty intervals). And this work has been published in Chen et al. (2014). However, we have noted that this IDA approach is suitable for incomplete data conditions, but when more data could be collected or when a continuous and random data distribution could be assumed, these intervals may not always be practical. Thus, we have proposed CDFA method.

The idea behind the CDFA was to replace the point-to-point comparison with the deviation between uncertain measured data and predicted data expressed as cumulative distribution functions. In fact, this is a modification of traditional good-of-fit indicators by replacing the calculations of their $O_i - P_i$ term by using stochastic distances between the paired probability density functions (PDFs). Thus, this CDFA could be used during the calibration and validation process if PDFs could be obtained for both prediction data and measurement data.

In other cases, the measurement and prediction uncertainties might be expressed as discrete data, or no continuous distribution function may fit the data set. For example, the input uncertainty relating to spatial rainfall variability might only result in a limited number of predicted data points that cannot be expressed as certain PDFs (Shen et al. 2012). Thus, we used MCA method. A key step is sampling variables randomly for discrete data so that the measurement and prediction data can be expressed as certain distributions. Here, ($O_i - P_i$) was replaced by a stochastic expression of the deviation between pairs of data groups, and these stochastic deviations were then used to calculate the evaluation indicators.

Based on the results obtained from this study, we found that the model performance worsened when a larger error range existed, and the choice of PDF affected the model performance, especially for non-point source (NPS) pollution

predictions. These proposed methods could be extended to other goodness-of-fit indictors and other watershed models to provide a substitution for traditional model evaluations within an uncertainty framework. Thus, the authors do believe our method could be a substitute of traditional goodness-of-fit indictors and they could be used for the calibration and validation process.

**Comments: 2)** Then, how can the proposed approaches be implemented within the calibration process to reduce model error?

**Response:** In this study, two methods, the Cumulative Distribution Function Approach (CDFA) and the Monte Carlo Approach (MCA), were proposed for different situations. Previous studies have noted that if prediction uncertainty exists, the predicted data could be expressed as a confidence interval (CI) or a probability density function (PDF) (Franz and Hogue, 2011; Shen et al., 2012). Current research tends to express uncertain data as certain function distributions to express an error term (Zhang et al., 2009), which might lead to a more feasible expression than the traditional indicators. However, prediction uncertainty can be expressed as discrete variables of interest. For example, uncertainty related to rainfall is currently recognized as the major challenge for hydrological modeling science. Many previous studies have investigated uncertainty associated with measurement errors and spatial variability associated with rainfall. A number of researchers have investigated deviations in measured data, such as Sun et al. (2000), Kavetski et al. (2006), Bárdossy and Das (2008), McMillan et al. (2011). This uncertainty originates mainly from inaccuracy in measuring devices, local meteorological effects, and errors in data transmission. This kind of uncertainty could be expressed as confidence interval (CI) or a probability density function (PDF).

However, another important source of uncertainty that can be expressed as discrete variables also existed. For example, even in ideal conditions, where the dense and well-distributed gauges are available, the rain-gauge network cannot fully capture every point over the watershed. It is more common to have only a few stations distributed in space over the watershed. Rainfall at unknown points is thus estimated by means of interpolation techniques. Several techniques—such as the Centroid method, the Thiessen Polygon method, IDW and the Kriging method—have been used in spatial interpolation to produce information on the spatial distribution of rainfall (Mamillapalli, 1998; Chaubey et al., 1999; Bárdossy and Das, 2008; Hamed et

al., 2009; Cho et al., 2009; Fu et al., 2011). It is therefore logical to take interpolation methods into account when determining the impacts of spatial rainfall variability on H/NPS predictions in large watershed.

Thus, the cumulative distribution functions were used to describe uncertain data in CDFA, and MCA was implemented using the Monte Carlo technique, which has been used in many hydrological uncertainty studies (Sun et al., 2008; Zhang et al., 2016). Here, $(O_i - P_i)$ was replaced by a stochastic expression of the deviation between pairs of data groups, and these stochastic deviations were then used to calculate the evaluation indicators. Actually, in H/WQ models, the process of the calibration is to filter the optimal parameters for the model. In some sense, the evaluation approach is used to determine if there are the optimal parameters. Based on the results in the paper, these two new approaches provided more accurate goodness-of-fit indicators for model evaluation compared to traditional methods. Thus, it can be considered that the traditional evaluation method can be replaced by CDFA or MCA to reduce model error.

However, we didn't incorporate the approaches with model because that there are some limitations in applying of the two approaches. 1) The study of measured and predicted uncertainty is few, thus the distributions of uncertainties are not ensured. 2) In CDFA and MCA methods, high computer performance is needed. Because, the ordinary computers won't support the mass data computation for calibration.

Thank you very much for your wonderful job. Hope that our responses are satisfactory. Best regards.

Best wishes,
Zhenyao Shen
Professor
School of Environment,
Beijing Normal University,
Beijing, China, 100875
Tel: +86-10-5880 0398

E-mail: zyshen@bnu.edu.cn

**Reference:**

Chen, L., Shen, Z., Yang, X., Liao, Q. and Yu, S.L., 2014. An Interval-Deviation Approach for hydrology and water quality model evaluation within an uncertainty framework. Journal of Hydrology, 509: 207-214.

Franz, K.J. and Hogue, T.S., 2011. Evaluating uncertainty estimates in hydrologic models: borrowing measures from the forecast verification community. Hydrology and Earth System Sciences, 15(11): 3367-3382.

Shen, Z.Y., Chen, L. and Chen, T., 2012. Analysis of parameter uncertainty in hydrological and sediment modeling using GLUE method: a case study of SWAT model applied to Three Gorges Reservoir Region, China. Hydrology and Earth System Sciences, 16(1): 121-132.

Zhang X, Liang F, Srinivasan R, et al, 2009. Estimating uncertainty of streamflow simulation using Bayesian neural networks[J]. Water Resources Research, 45(2):257-60.

Sun, X., Mein, R.G., Keenan, T.D., Elliott, J.F., 2000. Flood estimation using radar and raingauge data. Journal of Hydrology, 239, 4-18.

Kavetski, D., Kuczera, G., Franks, S.W., 2006. Bayesian analysis of input uncertainty in hydrological modeling: 1. Theory. Water Resources Research, 42, W03407.

Bárdossy, A., Das, T., 2008. Influence of rainfall observation network on model calibration and application. Hydrol. Earth Syst. Sc. 12, 77-89.

McMillan, H., Jackson, B., Clark, M., Kavetski, D., Woods, R., 2011. Rainfall uncertainty in hydrological modelling: An evaluation of multiplicative error models. Journal of Hydrology, 400, 83-94.

Mamillapalli, S. 1998. Effect of spatial variability on river basin stream flow modeling. PhD diss. West Lafayette, Ind.: Purdue University.

Chaubey, I., Haan, C.T., Grunwald, S., Salisbury, J.M., 1999. Uncertainty in the model parameters due to spatial variability of rainfall. Journal of Hydrology, 220, 48-61.

Hamed R., Patrickm W., Janm F., 2009. Effect of watershed delineation and areal rainfall distribution on runoff prediction using the SWAT model. Hydrology Research, 40, 505-519.

Cho, J., Bosch, D., Lowrance, R., Strickland, T., Vellidis, G., 2009. Effect of spatial distribution of rainfall on temporal and spatial uncertainty of SWAT output. Trans. ASABE 52, 1545-1555.

Fu, S.H., Sonnenborg, T.O., Jensen, K., He X., 2011. Impact of Precipitation Spatial Resolution on the Hydrological Response of an Integrated Distributed Water Resources Model. Vadose Zone J. 10, 25-36.

Shen, Z. et al., 2013. Uncertainty in flow and water quality measurement data: A case study in the Daning River watershed in the Three Gorges Reservoir region, China. Desalination and Water Treatment, 51(19-21): 3995-4001.

Sun, F., Chen, J., Tong, Q. and Zeng, S., 2008. Managing the performance risk of conventional waterworks in compliance with the natural organic matter regulation. Water Research, 42(1-2): 229-237.

Zhang, J., Li, Y., Huang, G., Chen, X. and Bao, A., 2016. Assessment of parameter uncertainty in hydrological model using a Markov-Chain-Monte-Carlo-based multilevel-factorial-analysis method. Journal of Hydrology, 538: 471-486.

---

## Author Response (AR2)

Dear Editor,

Thank you very much for your email of June 15, 2018, informing us of valuable suggestions to improve our manuscript 'Improvement of model evaluation by incorporating prediction and measurement uncertainty' (hess-2017-66). We wish to express our gratitude to the Hydrology and Earth System Sciences for encouraging us for the revised version of this paper, as well as to anonymous reviewers who provide valuable suggestions and professional revision of our manuscript. We have carefully considered your comments and revised the manuscript. The detailed responses are as follows:

===========================================================

**Response to the editor**

**1) Comments:** I would suggest to highlight methodological contribution to the literature in the manuscript.

**Response:** Thank you for the valuable suggestion. As suggested, we revised the manuscript accordingly. Please find the attached manuscript. In fact, the objective of this study is indeed to develop a new framework for model evaluation by incorporating prediction and measurement uncertainty. This methodological contribution is because in traditional indicators (such as Nash–Sutcliffe model efficiency), the deviation between the measured and predicted data is expressed by the absolute distance $(O_i - P_i)$ between the paired data points. This method is questionable because it fails to incorporate prediction and measurement uncertainty. Thus, the idea behind the CDFA was to replace the point-to-point comparison with the deviation between uncertain measured data and predicted data expressed as cumulative distribution functions. In fact, this is a modification of traditional good-of-fit indicators by replacing the calculations of their $O_i - P_i$ term by using stochastic distances between the paired probability density functions (PDFs). Thus, this CDFA could be used during the calibration and validation process if PDFs could be obtained for both prediction data and measurement data. Based on the results obtained from this study, we found that the model performance worsened when a larger error range existed, and the choice of PDF affected the model performance, especially for non-point source (NPS) pollution

predictions. These proposed methods could be extended to other goodness-of-fit indictors and other watershed models to provide a substitution for traditional model evaluations within an uncertainty framework. Thus, the authors do believe our method could be a substitute of traditional goodness-of-fit indictors and they could be used for the calibration and validation process in the future.

================================================================

**Response to the reviewer 2**

**1) Comments:** The proposed approach is more about a post calibration process for model evaluation rather than a new technique that can be implemented into the model during the calibration to gain knowledge from prediction and measurement uncertainty. This should be clarified in the manuscript. I would also suggest to comment the technical complexity in implementing this approach within the model (may be in the conclusion).

**Response:** As suggested, we revised the conclusion section in order to make clear description of this proposed approach. The revised conclusion is as follows:

"In this study, two new methods were proposed and employed to evaluate model performance within an uncertainty framework: the CDFA and the MCA. Using the CDFA and the MCA, both prediction and measurement uncertainty could be considered for model evaluation in a post calibration process, and the possible impacts of error range and the choice of PDFs could be quantified for a real application. Based on the results, the model performance worsened when a larger error range existed, and the choice of PDF affected the model performance, especially for NPS pollution predictions. These proposed methods could be extended to other goodness-of-fit indictors and other watershed models to provide a substitution for traditional model evaluations within an uncertainty framework. Thus, the new approaches could be a

substitute of traditional goodness-of-fit indictors and they could be used for the model evaluation process.

However, it should be noted the proposed CDFA and the MCA would serve for model evaluation in a post calibration process rather than a new calibration technique due to the technical complexity in implementing this approach within the model calibration. With the results presented, fixed PDFs or error range for prediction data could not be founded due to insufficient knowledge and natural randomness. Thus modellers should better assess the error range of measured data for their use in watershed simulations, and more data should be gathered to obtain a real measurement error range and a proper PDF for the predicted data. Further explanations are also suggested for the inherent uncertainty of hydrological and pollutant transportation processes. More case studies should be conducted to test the IDA, CDFA and MCA in future practical analyses of other watershed models."

Thank you very much for your wonderful job. Hope that our responses are satisfactory. Best regards.

Zhenyao Shen

Professor

School of Environment,

Beijing Normal University,

Beijing, China, 100875

Tel: +86-10-5880 0398

E-mail: zyshen@bnu.edu.cn

---

## Author Response (AR3)

Dear Editor,

Thank you very much for your email of July 24, 2018, informing us of valuable suggestions to improve our manuscript 'Improvement of model evaluation by incorporating prediction and measurement uncertainty' (hess-2017-66). We wish to express our gratitude to the Hydrology and Earth System Sciences for encouraging us for the revised version of this paper, as well as to anonymous reviewers who provide valuable suggestions and professional revision of our manuscript. We have carefully considered your comments and revised the manuscript. The detailed responses are as follows:

========================================================

**Response to the editor**

**1) Comments:** Given the history of the manuscript and on the basis of my own assessment, I do think the study can be accepted. I do recommend careful checking of the English usage, though, before final publication.

**Response:** Thank you for the valuable suggestion. As suggested, we revised the manuscript accordingly. Please find the attached manuscript.

Thank you very much for your wonderful job. Hope that our responses are satisfactory. Best regards.

Zhenyao Shen

Professor

School of Environment,

Beijing Normal University,

Beijing, China, 100875

Tel: +86-10-5880 0398

E-mail: zyshen@bnu.edu.cn